# Exploring OR2H1-Mediated Sperm Chemotaxis: Development and Application of a Novel Microfluidic Device

**DOI:** 10.3390/cells14130944

**Published:** 2025-06-20

**Authors:** Fiorella Di Nicuolo, Emanuela Teveroni, Alessandro Devigili, Clelia Gasparini, Andrea Urbani, Tullio Ghi, Alfredo Pontecorvi, Domenico Milardi, Francesca Mancini

**Affiliations:** 1International Scientific Institute Paul VI, Fondazione Policlinico Universitario A. Gemelli IRCCS, 00168 Rome, Italy; fiorella.dinicuolo@policlinicogemelli.it (F.D.N.); tullio.ghi@unicatt.it (T.G.); francesca.mancini@policlinicogemelli.it (F.M.); 2Department of Basic Biotechnological Sciences, Intensive and Perioperative Clinics, Catholic University of Sacred Heart, 00168 Rome, Italy; emanuela.teveroni@guest.policlinicogemelli.it (E.T.); andrea.urbani@policlinicogemelli.it (A.U.); 3Department of Biology, University of Padova, 35122 Padua, Italy; alessandro.devigili@unipd.it (A.D.); clelia.gasparini@unipd.it (C.G.); 4Fondazione Policlinico Universitario A. Gemelli IRCCS, 00168 Rome, Italy; 5Department of Life Sciences and Public Health, Catholic University of Sacred Heart Rome Campus, Fondazione Policlinico Universitario A. Gemelli IRCCS, 00168 Rome, Italy; 6Department of Women and Child Health, Fondazione Policlinico Universitario A. Gemelli IRCCS, 00168 Rome, Italy; 7Complex Operative Unit of Internal Medicine, Endocrinology and Diabetology, Department of Translational Medicine and Surgery, Fondazione Policlinico Universitario A. Gemelli IRCCS, 00168 Rome, Italy; alfredo.pontecorvi@policlinicogemelli.it

**Keywords:** olfactory receptor, OR2H1, chemotaxis, microfluidic device, methional, *Lactococcus lactis*

## Abstract

Microfluidic platforms have emerged as critical technologies for exploring sperm chemotaxis, providing precise gradient control, and facilitating in-depth behavioral assessment. We designed a novel, user-friendly microfluidic device that is optimized for human sperm morphology and motility. The device was validated using two well-established sperm chemoattractants, progesterone and bourgeonal, demonstrating its reliability and reproducibility. Given the key role of olfactory receptors (ORs) in mediating sperm chemotaxis, the newly developed device was employed to identify additional receptors that may contribute to sperm behavior. Using the Atlas database, we identified OR2H1 as a candidate receptor. It is enriched in testis-derived cells, particularly in early and late spermatids, and it is broadly expressed across human spermatozoa. We demonstrated that OR2H1’s ligand, methional, a sulfur-containing aldehyde naturally found in vaginal fluid and biosynthesized by *Lactococcus lactis*, significantly enhances sperm migration and progressive motility. Methional stimulation also triggered increased intracellular calcium levels, indicating receptor activation. Computer-assisted sperm analysis revealed that methional treatment improved sperm linearity, straightness, and wobble without affecting the average velocity, suggesting enhanced directional movement. These findings provide evidence that methional promotes sperm chemotaxis via OR2H1 and highlight the potential role of the vaginal microbiome in influencing human fertility.

## 1. Introduction

Microfluidic devices have become essential tools for studying sperm chemotaxis, as they offer precise control over chemical gradients and enable detailed analysis of sperm behavior. These devices allow researchers to isolate and quantify sperm chemotactic responses, enhancing both fundamental understanding and clinical applications in reproductive technology [1,2]. Engineered to create stable spatially and temporally controlled chemical gradients, microfluidic devices are crucial for accurately measuring sperm chemotaxis [1,3,4,5,6,7,8]. Designs often incorporate flow-through or flow-free configurations to distinguish chemotaxis from other behaviors, such as rheotaxis (movement in response to fluid flow) and thermotaxis (movement in response to temperature gradients) [1,3,4,5,6,7,8]. Some devices integrate both chemotaxis and thermotaxis assays, facilitating the simultaneous study of multiple guidance mechanisms [3,6,7]. These platforms have been used to test sperm responses to various chemo-attractants, including ovary or oviductal extracts and specific molecules like Acetylcholine and progesterone [1,4,5,6,7,8]. Microfluidic chips can quantify the proportion of sperm exhibiting chemotactic behavior; studies report that approximately 10% of sperm are chemotactically responsive under certain conditions [1,9]. Overcoming limitations of traditional assays, such as poor gradient stability and the inability to track single sperm over time, microfluidic devices enable high-throughput, parallel experiments and reduce experimental errors, making them suitable for both research and clinical sperm selection [4,7,8,10].

Microfluidic devices proved to be valuable tools to assess the role of olfactory receptors (ORs) in chemotaxis. Olfactory receptors are G-protein-coupled receptors that detect volatile chemicals acting as odorants [11]. OR activation initiates an intracellular signal transduction pathway triggered by adenylate cyclase III stimulation, leading to subsequent Ca^2+^ influx and membrane depolarization [12]. Initially detected in the olfactory epithelium, ORs have since been identified in various tissues, including the brain, bladder, prostate, and testis [11,13,14,15,16,17,18,19]. Notably, several studies indicate that the testis exhibits the richest OR expression outside of the nose [16,19,20]. Late spermatids and spermatozoa display selectively distinct OR expression across anatomical compartments, suggesting their potential roles in processes like maturation, migration, and fertilization [19,20,21,22,23]. Spermatozoa chemotaxis is largely dependent on intracellular calcium release, and ORs appear to be among the receptors involved in activating this signaling pathway [20,21,22,24,25]. The growing body of literature supports a close relationship between ORs and spermatozoa chemotaxis. For example, OR17-4 (also known as OR1D2) is located in the spermatozoon midpiece [21]. Spehr and colleagues discovered bourgeonal, a ligand for human OR17-4, which increases Ca^2+^ and induces chemotaxis in human spermatozoa [24]. Although bourgeonal is likely not an endogenous compound in the human body, this study suggested that human ORs function as chemoreceptors for small molecules in sperm [24]. Interestingly, reduced olfactory perception of bourgeonal has been correlated with idiopathic infertility [26]. Other odorants, such as Myrac (ligands of OR7A5), have also been shown to influence spermatozoa motility. This receptor appears to be activated by ligands found in vaginal secretions and follicular fluid, particularly 4-hydroxy-2,5-dimethyl-3[2H]-furanone [27]. Vaginal fluids contain a plethora of molecules that could potentially act as chemoattractants for spermatozoa. Among them, we recently demonstrated that propionate, a short-chain fatty acid, regulates sperm chemotaxis by activating its receptor OR51E2 [28]. We showed that propionate induces Ca^2+^ influx in sperm cells. Moreover, computer-assisted sperm analysis (CASA) demonstrated increased sperm linearity (LIN) and straightness (STR), indicating that propionate promotes linear sperm movement and functions as a chemoattractant via OR51E2 activation [28].

The primary aim of this research was the design of a new and accessible sperm device for chemotaxis studies, with a focus on creating a device that would be easy to use and provide reproducible results. This device was based on a modified version of the Devigili et al. chamber [29], with its shape and dimensions optimized for human sperm morphology and motility. To validate the consistency and reliability of the device, we used two well-established sperm chemo-attractants, progesterone [30,31] and bourgeonal [24].

Given the importance of olfactory receptors in sperm chemotaxis, we used the newly designed device to identify other receptors on spermatozoa that could function in sperm chemotactic response. We then queried the Atlas database to search for candidate ORs [32,33]. An analysis of 245 genes within the testis tissue expression cluster (spermatid development) revealed four ORs: OR10J1, OR4N4, OR14A2, and OR2H1 [32]. Furthermore, an Atlas analysis of 116 genes in the single-cell expression cluster of spermatid transcription identified three ORs: OR13A1, OR51I1, and OR2H1 [33]. Of these, only OR2H1 was enriched in testis-derived cells, particularly in early and late spermatids, suggesting its potential role also in sperm differentiation. Accordingly, several olfactory receptors are actively expressed during sperm differentiation and maturation, and their reduced expression in spermatocyte maturation arrest indicates a role in normal sperm development [19].

The OR2H1 protein shows a widespread distribution in human spermatozoa, localized to the acrosome cap, the caudal sperm head, and the midpiece, and appearing as punctate signals throughout the flagellum [20]. Moreover, Flegel and colleagues showed that methional is a ligand for OR2H1, capable of evoking Ca^2+^ signals in human spermatozoa [20]. Methional is a sulfur-containing aldehyde notable for its roles in food flavor, aroma, and cellular processes. It is primarily produced from methionine via the Strecker degradation reaction, which can occur during food processing or fermentation [34]. Notably, methional is a naturally occurring odorant in vaginal fluid and is biosynthesized by certain bacterial strains, including *Lactococcus lactis* [27,34,35]. These bacteria, naturally present in the vaginal microbiota, are under investigation for their probiotic benefits in vaginal health, such as antimicrobial and immune-modulating activities [36,37]. Given the significance of the OR2H1 receptor and its specific ligand—an endogenous compound produced by the vaginal microbiota within the female reproductive tract—we aimed to elucidate the involvement of their interaction in sperm chemotaxis and its potential implications for fertility.

## 2. Materials and Methods

### 2.1. Sperm Samples: Ethical Approval and Biological Preparation

Sperm collection and analysis were conducted with approval from the local ethics committee of the Fondazione Policlinico Universitario A. Gemelli, IRCCS, Rome, Italy (protocol number ID 3943; date of approval 15 April 2021). Fresh human sperm were obtained via masturbation from young (age range 20–40 years old) healthy donors (3–5 days abstinence) who provided informed, signed consent. A comprehensive evaluation was performed, examining seminal parameters such as the pH, volume (mL), sperm count (10^6^/mL), total sperm count (10^6^/ejaculate), total and progressive motility (%), morphology (percentage of normal and abnormal forms), and leukocyte levels (10^6^/mL). All samples are in accordance with the WHO 2021 semen-quality criteria for sperm concentration (>15 million/mL), motility (>40% motile cells), and viability (>58% live cells). Following liquefaction at 37 °C for 30 min, the semen samples were prepared for experiments. For device migration assays, sperm were pre-incubated in a capacitation medium (1:1 in HAM1x supplemented with 0.6 mg/mL BSA and 0.2 mg/mL NaHCO_3_-Sigma-Aldrich, St. Louis, MO, USA) for 1 h at 37 °C. For the computer-assisted sperm analysis (CASA, Microptic S.L., Barcelona, Spain) of motility, semen samples were diluted in a capacitation medium (1:10) with or without 10 µM methional (Sigma-Aldrich, St. Louis, MO, USA)and analyzed after 1 and 3 h of incubation.

### 2.2. Design and Validation of the Sperm Device

We designed a sperm device based on a modification of the device previously described by Devigili et al. [29], with its shape and dimensions tailored to human sperm morphology and motility. In particular, given that rapidly motile sperm travel at least 10–25 μm per second in a relatively straight path [38], we estimated that a 2.5 cm channel length would allow sperm to accumulate in the recovery wells within 15–30 min of incubation.

The device (Figure 1) consists of a central well connected to three separate channels, each ending in a well. Detailed specifications of the device, including the size and depth of the wells and channels, are available in the Appendix A, along with a 3D-printable file. This file enables direct fabrication of the device or its adaptation for specific needs, such as altering channel lengths. The 3D model (Appendix A) was developed using Autodesk^®^ Fusion360 (San Rafael, CA, USA). The physical devices were 3D printed with high precision (29-micron accuracy) in a biocompatible resin (VisiJet^®^ Crystal, EX 200 Plastic Material, USP Class VI certified for medical applications; 3D Systems, Rock Hill, SC, USA) using the lost-wax method. The central well was filled with the capacitating solution, and a chemoattractant was added to one of the terminal wells (A or B). This creates a concentration gradient of the chemoattractant between the well and the central well, whereas the other channel serves as a control without any gradient. Visual observation of dye diffusion in the central well indicated that a 20 s interval was sufficient for gradient formation (Appendix A). Then, we monitored the stability of the gradient: it remained stable for at least 30 min (Appendix A).

Sperm were introduced into the third terminal well (S) and allowed to swim into the central well, where they underwent capacitation. Sperm that exhibited chemotaxis towards the chemoattractant well (B) or that randomly entered the control well (A) were collected via pipetting.

The operational protocol for the sperm choice device is as follows:➢Priming: Introduce 100 μL of capacitation medium (HAM1x supplemented with 0.6 mg/mL BSA and 0.2 mg/mL NaHCO_3_-Sigma-Aldrich, St. Louis, MO, USA) into the three peripheral wells (S, A, and B). This step primes the device, filling the central well and eliminating air pockets within the channels.➢Gradient Formation: Add 10 μL of the designated chemoattractant solution to well B and 10 μL of the control solution (capacitation medium) to well A. Allow a 20 s period for the establishment of a chemoattractant gradient, extending from each well through its respective channel (channel B or A) and into the central well.➢Sperm Introduction: Carefully introduce 40 μL of sperm pre-incubated in a capacitation medium into the designated sperm well, S.➢Sperm Collection and Analysis: Following a 30 min incubation at 37 °C, collect 20 μL aliquots of migrated sperm from the two collection wells (B and A). Quantify the collected sperm using an optical microscope.

A schematic representation of the operational protocol is shown in Appendix A. Sperm cell numbers were evaluated by a Makler Chamber in accordance with the WHO 2021 recommendations. To validate the reliability of the sperm choice device as a research tool, we tested two well-known sperm chemo-attractants (5 nM progesterone, Prontogest^®^, IBSA Farmaceutici srl, Lodi, Italy, and 1 µM bourgeonal, Sigma-Aldrich, St. Louis, MO, USA). To this end, five different sperm ejaculates were tested, as described above.

### 2.3. Analysis of OR2H1 from Atlas Database

OR2H1’s RNA expression was analyzed using the Atlas database (https://www.proteinatlas.org/ENSG00000204688-OR2H1 (accessed on 10 March 2024). This database provides RNA-seq data from two sources: the internally generated Human Protein Atlas (HPA) and the Genotype-Tissue Expression (GTEx) project, along with a consensus dataset combining both. The consensus dataset presents normalized expression levels (nTPM, normalized transcripts per million) across 55 tissue types. RNA expression data were examined at both the tissue expression level (https://www.proteinatlas.org/ENSG00000204688-OR2H1/tissue#rna_expression (accessed on 10 March 2024) and the single-cell expression level (https://www.proteinatlas.org/ENSG00000204688-OR2H1/single+cell (accessed on 10 March 2024).

### 2.4. Measurement of Calcium Changes

Intracellular free calcium concentration (Ca^2+^) changes in human sperm cells were measured using a fluorescence plate reader (Varioskan Lux, Thermo Fisher, Waltham, MA, USA) in 96-well plates. Briefly, seminal fluid aliquots (containing ≅ 10 × 10^6^ sperm cells) were incubated with 10 μM of the fluorescent Ca^2+^ indicator Fluo-4, AM (Fluo-4 Direct Calcium Assay kit, Thermo Fisher, Waltham, MA, USA) for 30 min at 37 °C. Fluorescence was excited at 480 nm, and emissions were recorded at 520 nm using bottom optics. Measurements were taken before and after the addition of the test compound (10 µM methional- Sigma-Aldrich, St. Louis, MO, USA) and controls in duplicate wells. Changes in Fluo-4 fluorescence were expressed as relative fluorescent signals (RFSs), calculated with respect to the basal fluorescence before compound addition.

### 2.5. Evaluation of Sperm Motility Parameters by CASA

Sperm motility was analyzed using a computer-assisted sperm analysis (CASA) system (Microptic S.L., Barcelona, Spain). Briefly, semen samples were diluted in the capacitation medium, and for each treatment group (with or without 10 µM methional incubation for 1 and 3 h), several motility parameters of spermatozoa were assessed as follows: the curvilinear velocity (VCL, μm/s), the straight-line velocity (VSL, μm/s), the path velocity (VAP, μm/s), the beat cross frequency (BCF, Hz), the amplitude lateral head displacement (ALH, μm), the straightness (STR, %), the linearity (LIN, %), and the wobble index (WOB, %). Then, 3 μL of each sperm sample was loaded onto a pre-warmed (37 °C) standard-count four-chamber Leja slide (Leja Products BV, The Netherlands, cod. SC 20-01-04-B). A minimum of 500 sperm cells were examined across at least five different fields per group.

### 2.6. Statistical Analysis

The results are presented as the mean ± standard deviation (SD). Statistical significance was determined using paired two-tailed Student’s *t*-tests or one-sample *t*-tests (when the control group mean was normalized to one), with a significance level set at *p* < 0.05. GraphPad Prism 7.04 software was used for all statistical analyses.

## 3. Results

### 3.1. Sperm Device Development and Validation

To study sperm chemotaxis, a custom-designed sperm device was developed. The device features a central well connected to three terminal wells via channels (Figure 1). A capacitating solution fills the central well, and a chemoattractant is added to the terminal well (B) to create a concentration gradient, while the other serves as a control (A). Sperm are introduced through a third well (S) and allowed to swim into the central chamber. The sperm derived from chemo-attractant or control channels were collected for analysis. A technical drawing of the sperm chamber showing the key structural dimensions is shown in Appendix A. The 3D-printable file is also available in the Appendix A.

**Figure 1 cells-14-00944-f001:**
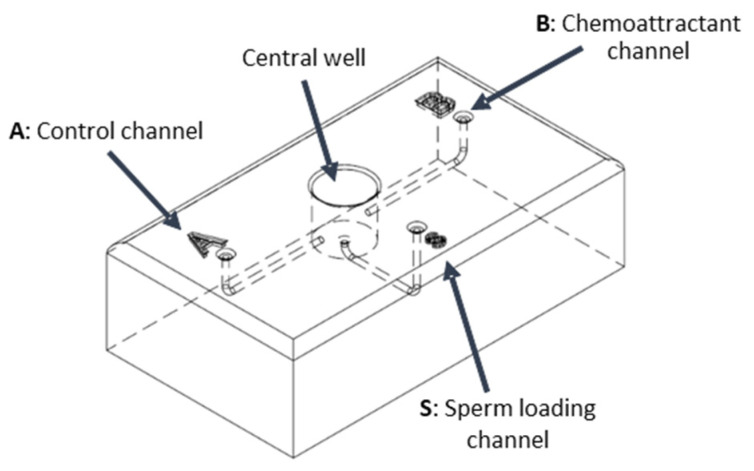
Schematic representation of the sperm device. The central well serves as the main site for sperm activation and swimming. Well S is used for sperm loading. Wells A and B lead to the respective channels and are designated to introduce the testing solutions (the control and the chemoattractant solution, respectively, in our experiment). The chamber wells are labeled (A, B, S) to facilitate tracking the loading and collection procedures. Further details and a step-by-step protocol are provided in the Materials and Methods section. The ready-to-print 3D file is available in the Appendix A, as well as a detailed project of the sperm device.

To validate the sperm choice device’s reliability as a research tool, we tested two well-established sperm chemo-attractants, progesterone [30,31] and bourgeonal [24]. Figure 2A shows the migrated spermatozoa recovered in the well containing progesterone (P) with respect to the control well (CTR).

As expected, progesterone causes a significant increase in the number of recovered sperm. Moreover, as shown in Figure 2B, progesterone treatment leads to a significant enhancement of sperm progressive motility.

To confirm that observed sperm movement is due to a chemotactic response and not random motion or other factors, we performed the same experiment without progesterone (no gradient) or using a reverse gradient (where the chemoattractant is placed in the opposite well). As shown in Appendix A, when both wells contained the same solution (e.g., medium without chemoattractant), sperm movement was random and not directed by a chemical gradient. Moreover, Appendix A shows the migrated spermatozoa recovered in the well containing progesterone (P) with respect to the control well (CTR) using a reverse gradient.

The odorant bourgeonal was used to further validate the sperm device’s accuracy. Figure 2C demonstrates a significant increase in the number of migrated spermatozoa recovered in the bourgeonal-containing well (B) compared to the control well (CTR), as expected. Finally, bourgeonal treatment resulted in a significant enhancement of sperm progressive motility, as illustrated in Figure 2D. These experiments validate the sperm device as a reliable and accessible tool for studying human sperm chemotaxis.

### 3.2. Analysis of OR2H1 Expression

To study novel potential receptors involved in sperm chemotaxis, the Atlas database was used to search for olfactory receptors (ORs) in spermatozoa. The analysis revealed that the tissue RNA expression of OR2H1 (https://www.proteinatlas.org/ENSG00000204688-OR2H1/tissue#rna_expression (accessed on 10 March 2024) was enriched in the testis expression cluster composed of 55 tissues (Spermatid development) (Figure 3A). Additionally, the Atlas analysis of the single-cell-specific RNA expression cluster of the spermatid transcription identified OR2H1 (https://www.proteinatlas.org/ENSG00000204688-OR2H1/single+cell (accessed on 10 March 2024) as enriched in testis-derived cells, particularly in early and late spermatids across 81 cell types (Figure 3B).

### 3.3. Analysis of Methional’s Effect on Sperm Migration and Activation

To examine the consistency and reliability of the results obtained using the newly developed device, we tested it with the OR2H1 ligand, methional. Figure 4A shows the number of migrated spermatozoa recovered in the well containing methional (M) compared to the control well (CTR).

Methional significantly increased sperm recovery. To further validate our findings, we performed dose–response experiments (methional 10 and 100 µM). Appendix A shows that methional elicited sperm accumulation in a dose-dependent manner, supporting its role as a chemoattractant. For all subsequent experiments, we selected the lowest concentration of 10 µM, which may approximate the hypothetical physiological concentration found in the vaginal environment. We further analyzed methional’s effect on sperm motility. As shown in Figure 4B, methional treatment significantly enhanced the progressive motility of spermatozoa, comparable to the effects observed with bourgeonal and progesterone. To investigate OR2H1 signaling following methional stimulation, we evaluated the Ca^2+^ influx. Seminal fluid was loaded with a fluorescent Ca^2+^ indicator, and cells were rapidly stimulated with 10 μM methional. A real-time recording of the fluorescence signal demonstrated a significant increase in intracellular Ca^2+^ after acute methional injection compared to non-treated sperm (Figure 4C,D). These experiments strongly suggest that methional activates its receptor, stimulating sperm chemotaxis.

### 3.4. Analysis of Methional’s Effect on Sperm Kinematics Parameters

To further analyze methional’s effects on sperm motility, we used the CASA system to assess the kinematic parameters of motile sperm. We observed similar values for the curvilinear velocity (VCL, μm/s), the straight-line velocity (VSL, μm/s), the average path velocity (VAP, μm/s), the beat cross frequency (BCF, Hz), and the amplitude lateral head displacement (ALH, μm) after both one and three hours of methional treatment, compared to the control (CTR) (Figure 5A–E). Importantly, statistically significant increases were observed for several motility parameters, including the linearity (LIN %), straightness (STR, %), and wobble index (WOB, %), in sperm cells exposed to methional for one and three hours, compared to the control (CTR) (Figure 5F–H). Therefore, while the VCL, VSL, VAP, BCF, and ALH parameters were unaffected, the observed increases in LIN, STR, and WOB suggest that methional specifically enhanced the fraction of spermatozoa exhibiting straight, directional movement, in addition to its general effect on sperm migration.

Figure 6 shows a proposed mechanism by which methional, a naturally occurring odorant in the female reproductive tract, influences sperm behavior.

The proposed schematic representation illustrates that methional acts as a ligand for the ectopic olfactory receptor OR2H1 present in the sperm. This binding event triggers a modulation of calcium signaling within the sperm cell, inducing sperm chemotaxis.

## 4. Discussion

In the present work, we have successfully developed and validated a novel microfluidic sperm device based on a modified Devigili et al. design [29] optimized for human sperm characteristics. The device, with its precisely defined geometry and readily available 3D printable files, allows for the creation of stable chemoattractant gradients and the subsequent collection of sperm exhibiting chemotactic behavior. Our validation experiments used two well-established chemoattractants: progesterone and bourgeonal. Progesterone, secreted by cumulus cells, is a strong chemoattractant for mammalian sperm at low concentrations, guiding them towards the egg [39]. This process involves activating the adenylate cyclase pathway, protein tyrosine phosphorylation, and calcium mobilization, resulting in sperm orienting towards the progesterone source [31]. While progesterone can induce hyperactivation-like motility, directional chemotaxis is specifically driven by the progesterone gradient [40]. Bourgeonal activates the olfactory receptor hOR17-4 on human sperm, triggering a signaling cascade that is essential for chemotaxis [24]. This receptor is present both in the nose and sperm, indicating a shared mechanism for odor detection and sperm guidance [26].

The primary focus in developing the new sperm device for chemotaxis studies was to create a device that is both easy to operate and capable of producing consistent results. Our results demonstrated the device’s reliability in inducing and quantifying sperm chemotaxis in response to progesterone and bourgeonal. Specifically, we observed a significant increase in the number of migrated sperm towards the chemoattractant wells compared to the control, and importantly, as expected, these selected sperm displayed significantly enhanced progressive motility. These findings confirm the functionality of the novel sperm device as a valuable research tool for investigating sperm chemotaxis and for potentially isolating highly motile and functional spermatozoa for downstream applications.

To identify novel olfactory receptors (ORs) potentially guiding sperm chemotaxis, given the established importance of ORs in this process, we employed our newly designed device and queried the Atlas database. This analysis revealed several candidate ORs within the testis tissue expression cluster (spermatid development cluster: OR10J1, OR4N4, OR14A2, and OR2H1) and within the single-cell expression cluster of spermatid transcription (OR13A1, OR51I1, and OR2H1). Among them, OR2H1 was enriched in testis-derived cells, specifically in early and late spermatids, suggesting it may play a role in sperm differentiation as well as chemotaxis. This finding is supported by broader evidence showing that several olfactory receptors are actively expressed during normal sperm differentiation and maturation [19]. Having a functional role, the OR2H1 protein is widely present in human sperm, and its ligand, methional, a methionine-derived sulfur aldehyde, can trigger calcium signaling [20]. Of note, methional is a naturally occurring odorant in vaginal fluid [27]. Its enzymatic conversion by certain bacteria, such as *Lactococcus lactis*, involves aminotransferase and alpha-ketoacid decarboxylase activities, and this process is strain-dependent [34,35]. *Lactococcus lactis*, a lactic acid bacterium, is naturally present in the vaginal microbiota and has been studied for its probiotic properties and potential therapeutic applications in vaginal health. Several studies explored its role in antimicrobial activity, immune modulation, and as a candidate for probiotic therapies targeting vaginal and systemic conditions [36,37]. Moreover, symbiotic associations of *Lactococcus lactis* with lactobacilli enhance colonization, biomass accumulation, and antimicrobial activity, supporting recolonization and prevention of bacterial vaginosis, especially under fluctuating vaginal pH conditions [41].

Given the presence of OR2H1 and its endogenous ligand in the female reproductive tract, the ultimate aim of this study was to explore its role in sperm chemotaxis. Our results demonstrated that methional significantly increased the number of migrated sperm and enhanced their progressive motility. Furthermore, methional stimulation led to a significant increase in intracellular calcium in sperm, indicating receptor activation. The CASA analysis revealed that while the curvilinear velocity (VCL), the straight-line velocity (VSL), the average path velocity (VAP), the beat cross frequency (BCF), and the amplitude lateral head displacement (ALH) remained unchanged, methional treatment significantly increased linearity (LIN), straightness (STR), and wobble (WOB), suggesting that it promotes more directional sperm movement. Overall, we showed that methional was able to induce sperm chemotaxis by an interaction with its receptor, OR2H1. The proposed mechanism by which methional activates its receptor, OR2H1, and induces sperm chemotaxis involves the interaction of the receptor with a GTP-binding protein (Golf). This interaction triggers the release of the GTP-bound Gαolf subunit, which subsequently activates adenylyl cyclase III (ACIII), leading to an increased production of cyclic AMP (cAMP). Elevated cAMP levels are thought to open calcium channel (e.g., CatSper), resulting in a rise in intracellular Ca^2+^ concentration and membrane depolarization. This intracellular signalling cascade ultimately modulates sperm chemotaxis. While CatSper has been proposed as the calcium channel involved—given that it is the primary calcium channel in sperm—its activation by odorants remains unclear. CatSper is well-known to be directly activated by female-derived factors, such as progesterone and prostaglandins, which stimulate Ca^2+^ influx. Several studies have suggested that odorants may regulate calcium entry and sperm motility via G-protein-coupled receptors (GPCRs) and cAMP-dependent pathways [21,24,25]. However, evidence from Brenker et al. indicates that certain odorants can activate CatSper directly, independent of GPCR and cAMP signaling [42]. To date, only one study has demonstrated that the olfactory receptor OR1D2 mediates bourgeonal-induced CatSper activation through a G-protein-dependent mechanism [43]. Therefore, it will be of great interest to identify the specific calcium channel involved in olfactory receptor OR2H1 signaling in future studies.

The presence of methional in vaginal fluid, likely from *Lactococcus lactis*, underscores the microbiome’s importance in reproduction, notably in affecting sperm motility. The vaginal microbiome, particularly its composition and diversity, plays a significant role in women’s reproductive health and fertility. Recent research highlights the importance of a Lactobacillus-dominated vaginal environment for optimal fertility and successful pregnancy outcomes, both naturally and in assisted reproductive technologies (ART) [44]. Moreover, the vaginal microbiome composition has been shown to influence sperm function and fertility [45]. This recent research explores how specific bacterial populations in the vagina can affect sperm motility and chemotaxis, potentially affecting conception rates and reproductive health. Couples, where the female partner had an altered vaginal microbiome, showed significantly lower sperm progressive motility compared to those with a normal microbiome. Sperm concentration and morphology were also lower in these couples, though not statistically significant [45]. The relative abundance of certain bacteria, including *Lactobacillus*, *Alkalibacillus*, *Streptococcus*, *Propionibacterium*, *Chitinophaga*, *Escherichia*, *Enterococcus*, and *Xenophylus*, explained a substantial portion of the variability in sperm motility [45]. The present study directs attention to the significant role of a specific component of the microbiome, *Lactococcus lactis*, and its metabolic product, methional, in modulating sperm motility and ultimately affecting fertility. By focusing on the interplay between this bacterium, its secreted molecule, and sperm physiology, we highlight a novel, potentially crucial link between the vaginal microbiome and reproductive success. Understanding this interaction could reveal novel pathways that influence sperm behaviour and offer new perspectives on factors affecting fertility.

## 5. Conclusions

This study developed and validated a user-friendly and cost-effective microfluidic device for reliably studying sperm chemotaxis. This device is both easy to operate and capable of producing consistent results. Our research, using the novel device, investigates the role of chemosensory receptors in sperm behaviour, particularly OR2H1’s response to methional, a vaginal fluid compound. This finding also highlights a potential link between the vaginal microbiome, specifically *Lactococcus lactis* and its metabolite methional, and sperm motility, suggesting a novel pathway influencing fertility.

## Figures and Tables

**Figure 2 cells-14-00944-f002:**
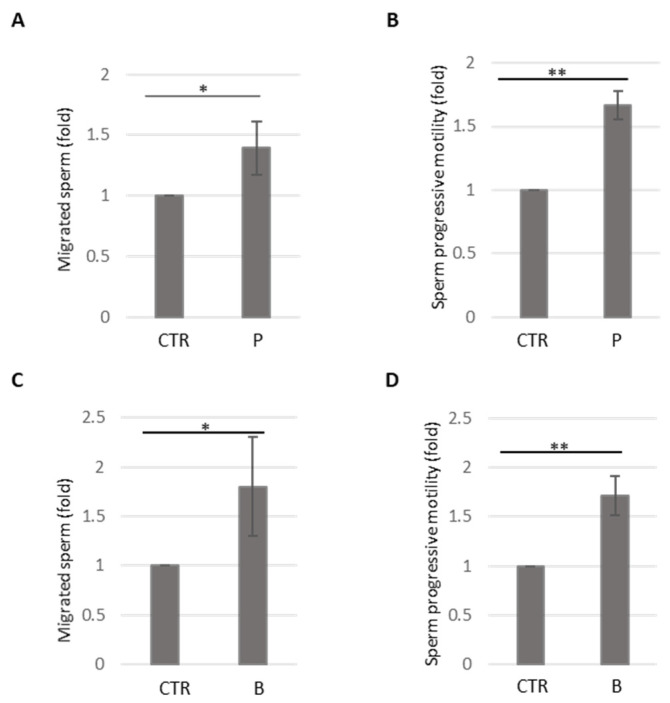
(**A**) The histogram shows the migrated spermatozoa recovered in well A (CTR) and well B (P, progesterone). Data are presented as the fold of recovered sperm upon progesterone treatment relative to the CTR untreated well. The number of sperm cells (CTR) recovered in the untreated well was arbitrarily set to 1. The mean ± SDs of five independent experiments are shown (*N* = 5, * *p* < 0.05, one-sample *t*-test). (**B**) The histogram shows the progressive motility of spermatozoa recovered in well A (CTR) and well B (P, progesterone). Data are presented as the fold of recovered sperm progressive motility upon progesterone treatment relative to the CTR untreated well. The motility of sperm cells (CTR) recovered in the untreated well is arbitrarily set to 1. The mean ± SDs of five independent experiments are shown (*N* = 5, ** *p* < 0.01, one-sample *t*-test). (**C**) The histogram shows the migrated spermatozoa recovered in well A (CTR) and well B (B, bourgeonal). Data are presented as the fold of recovered sperm upon bourgeonal treatment relative to the CTR untreated well. The number of sperm cells (CTR) recovered in the untreated well is arbitrarily set to 1. The mean ± SDs of five independent experiments are shown (*N* = 5, * *p* < 0.05, one-sample *t*-test). (**D**) The histogram shows the progressive motility of spermatozoa recovered in well A (CTR) and well B (B, bourgeonal). Data are presented as the fold of recovered sperm progressive motility upon bourgeonal treatment relative to the CTR untreated well. The motility of sperm cells (CTR) recovered in the untreated well is arbitrarily set to 1. The mean ± SDs of five independent experiments are shown (*N* = 5, ** *p* < 0.01, one-sample *t*-test).

**Figure 3 cells-14-00944-f003:**
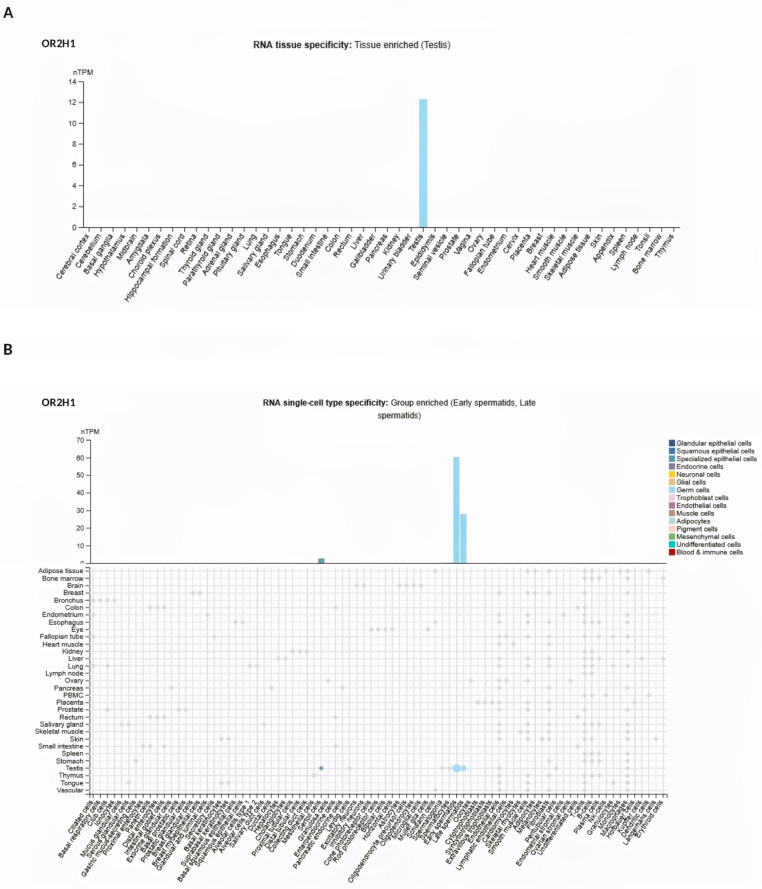
(**A**) OR2H1 tissue-specific enriched RNA expression in the Atlas database (https://www.proteinatlas.org/ENSG00000204688-OR2H1/tissue#rna_expression; accessed on 10 March 2024). The consensus dataset presents normalized expression levels (nTPM, normalized transcripts per million) across 55 tissue types. (**B**) OR2H1 single-cell type specific RNA expression in the Atlas database (https://www.proteinatlas.org/ENSG00000204688-OR2H1/single+cell; accessed on 10 March 2024). The consensus dataset presents normalized expression levels (nTPM) across 81 cell types.

**Figure 4 cells-14-00944-f004:**
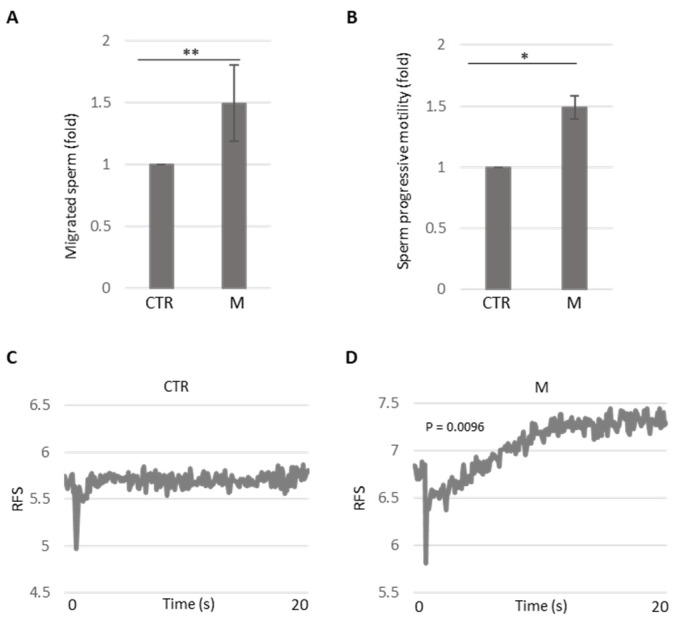
(**A**) The histogram shows the migrated spermatozoa recovered in well A (CTR) and well B (M, methional 10 µM). Data are presented as the fold of recovered sperm upon methional treatment relative to the CTR untreated well. The number of sperm cells (CTR) recovered in the untreated well is arbitrarily set to 1. The mean ± SDs of five independent experiments are shown (*N* = 5, ** *p* < 0.01, one-sample *t*-test). (**B**) The histogram shows the progressive motility of spermatozoa recovered in well A (CTR) and well B (M, methional). Data are presented as the fold of recovered sperm progressive motility upon methional treatment relative to the CTR untreated well. The motility of sperm cells (CTR) recovered in the untreated well is arbitrarily set to 1. The mean ± SDs of five independent experiments are shown (*N* = 5, * *p* < 0.05, one-sample *t*-test). (**C**,**D**) Time–course curve of Ca^2+^ influx in seminal fluids treated with PBS (CTR) or methional (M). The fluorescent signals recorded at 0 (starting time point) and 20 s (final time point) are obtained for statistical analysis in comparison to the untreated samples (CTR). The mean ± SDs of three independent experiments are shown (*N* = 3, *p*-value = 0.0096, one-sample *t*-test).

**Figure 5 cells-14-00944-f005:**
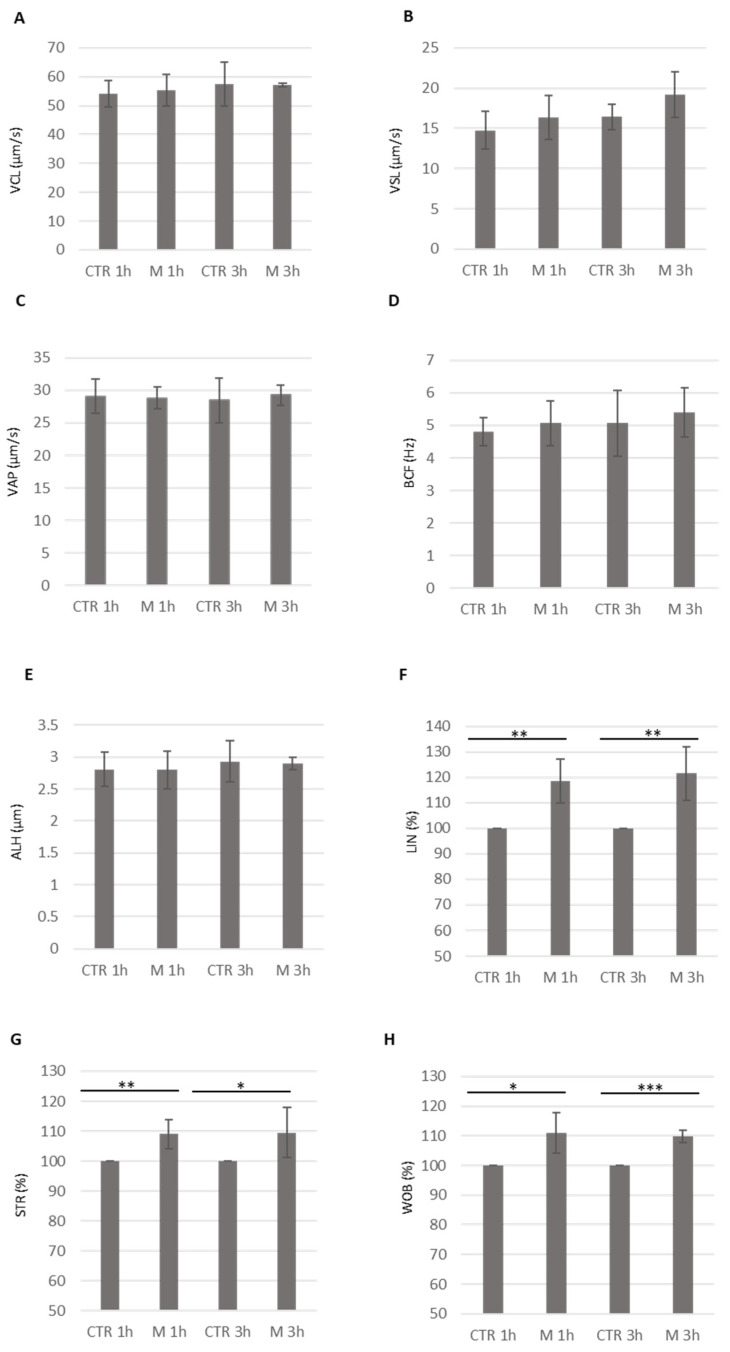
The histograms show the kinematic parameters obtained by the CASA system relative to the sperm cells incubated in the control medium (CTR) or a medium containing methional for 1 and 3 h at 37 °C. In (**A**), the average curvilinear velocity (VCL, μm/s) is shown; in (**B**), the average straight-line velocity (VSL, μm/s) is shown; in (**C**), the average path velocity (VAP, μm/s) is shown; in (**D**), the average beat cross frequency (BCF, Hz) is shown; in (**E**), the average amplitude lateral head displacement (ALH, μm) is shown; in (**F**), the linearity parameter (LIN, %) is shown; in (**G**), the straightness parameter (STR, %) is shown; in (**H**), the wobble index parameter (WOB, %) is shown. In (**F**–**H**), the kinematic values are represented as the percentage of methional (M)-treated cells with respect to untreated sperm (CTR, arbitrarily set to 100). The mean ± SDs of 5 independent experiments are shown (*N* = 5. * *p* < 0.05, ** *p* < 0.01, *** *p* < 0.001, one-sample *t*-test).

**Figure 6 cells-14-00944-f006:**
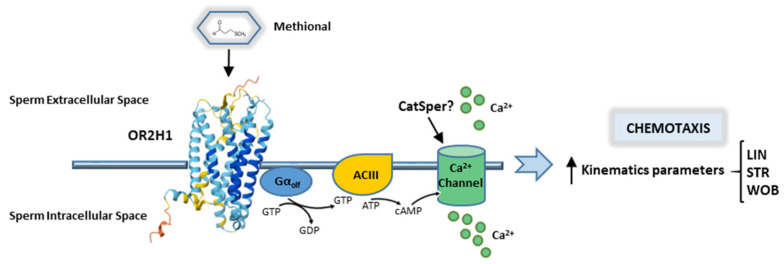
Proposed schematic pathway of ectopic OR2H1 in sperm: the binding of methional to the OR2H1 receptor leads to the interaction of the receptor with a GTP-binding protein, Golf. This interaction, in turn, leads to the release of a GTP-coupled Gαolf subunit, which then stimulates adenylyl cyclase III (ACIII) to produce elevated levels of cyclic AMP (cAMP). The increase in cyclic AMP opens the Ca^2+^ channel (CatSper?), leading to an increase in intracellular Ca^2+^ concentration and depolarization of the cell membrane. This intracellular cascade modulates sperm chemotaxis. LIN: the linearity parameter; STR: the straightness parameter; WOB: the wobble index parameter.

## Data Availability

The original contributions presented in this study are included in the article/Appendix A. Further inquiries can be directed to the corresponding author.

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
