# Peer review of "Exploring OR2H1-Mediated Sperm Chemotaxis: Development and Application of a Novel Microfluidic Device"

_cells, 2025, doi:10.3390/cells14130944_

Round 1
Reviewer 1 Report
Comments and Suggestions for Authors
This study develop a microfluidic device to study the sperm chemotaxis. OR2H1 was chosen as a candidate receptor to meditate sperm chemotaxis. The approach proved that Methional - a naturally occurring odorant in vaginal fluid enhanced sperm chemotaxis via OR2H1. This is exciting research with well-written Abstract, sufficient background in Introduction, appropriate method and design, and clear result and conclusion. The manuscript is suitable for the journal. The following comments may help the authors to improve the manuscript before acceptance.
- The authors claimed that they employed "novel microfluidic device" to study sperm chemotaxis. Specific ideas should be added to the manuscript to clarify the significance of this design in comparison with Devigili et al. design.
- A schematic of device operation should be added.
Author Response
This study develop a microfluidic device to study the sperm chemotaxis. OR2H1 was chosen as a candidate receptor to meditate sperm chemotaxis. The approach proved that Methional - a naturally occurring odorant in vaginal fluid enhanced sperm chemotaxis via OR2H1. This is exciting research with well-written Abstract, sufficient background in Introduction, appropriate method and design, and clear result and conclusion. The manuscript is suitable for the journal. The following comments may help the authors to improve the manuscript before acceptance.
- The authors claimed that they employed "novel microfluidic device" to study sperm chemotaxis. Specific ideas should be added to the manuscript to clarify the significance of this design in comparison with Devigili et al. design.
Thank you for the comment. As we mentioned in the paper, we modified the chamber previously described by Devigili et al., with shape and dimensions tailored to human sperm morphology and motility. Given that rapidly motile sperm travel at least 10-25 micrometers per second in a relatively straight path, we estimated a 2.5 cm channel length would allow sperm to accumulate in the recovery channels within 15-30 minutes of incubation. We added this information in the text.
2. A schematic of device operation should be added.
Thanks for your suggestion. We added a scheme of device operations in supplementary figure 3.
Reviewer 2 Report
Comments and Suggestions for Authors
The manuscript entitled : Exploring OR2H1-Mediated Sperm Chemotaxis: Development 2 and Application of a Novel Microfluidic Device is interesting and worth to be published after Minor Revision. The document is about practical sound and it shows OR2H1- mediated sperm chemotaxis event.
My concern is only about a few things:
- Is this possible to show for reader other sperm motion parameters? Such as total TMOT [%], PMOT [%], VSL [µm/s], VCL [µm/s, ALH [µm], BCF [Hz], Rapid [%], Medium [%], Slow [%], Static [%]. Why do you present only VAP, STR, LIN and WOB?
- Please use in all document one name for you Device also in figure captions (Fig.1).
- In all document use mL except of ml
- Line 206. Did you use seminal fluid aliquots or semen? Were there any sperm?
- Figure 5 need to be improved. Please remove from y axis description “Increase of kinematic parameter”. Leave only parameter name and units. Or maybe you should prepare table with all sperm motion parameters?
- Figure 6. Be precise and add that it is in human sperm.
- Line 380. Where do you show significantly enhanced progressive motility in this paper?
Author Response
The manuscript entitled : Exploring OR2H1-Mediated Sperm Chemotaxis: Development 2 and Application of a Novel Microfluidic Device is interesting and worth to be published after Minor Revision. The document is about practical sound and it shows OR2H1- mediated sperm chemotaxis event.
My concern is only about a few things:
- Is this possible to show for reader other sperm motion parameters? Such as total TMOT [%], PMOT [%], VSL [µm/s], VCL [µm/s, ALH [µm], BCF [Hz], Rapid [%], Medium [%], Slow [%], Static [%]. Why do you present only VAP, STR, LIN and WOB?
Thanks for your suggestion. We correct the figure 5 adding several sperm motion parameters.
2. Please use in all document one name for you Device also in figure captions (Fig.1).
Thanks for your suggestion. We correct it in the text.
3. In all document use mL except of ml
Thanks for your suggestion. We correct it in the text.
4. Line 206. Did you use seminal fluid aliquots or semen? Were there any sperm?
Thanks for your comment. For seminal fluid we mean total semen fluid contains sperm cells. Anyway, we added in the text this information (seminal fluid aliquots containing ≅10x106 sperm cells).
5. Figure 5 need to be improved. Please remove from y axis description “Increase of kinematic parameter”. Leave only parameter name and units. Or maybe you should prepare table with all sperm motion parameters?
Thanks for your suggestion. We correct the figure 5.
6. Figure 6. Be precise and add that it is in human sperm.
Thanks for your suggestion. We added this information in the legend.
7. Line 380. Where do you show significantly enhanced progressive motility in this paper?
Thanks for your comment. We showed these data in figure 2B, and 2D.
Reviewer 3 Report
Comments and Suggestions for Authors
In this manuscript, from Di Nicuolo and colleagues, the authors present the development of an accessible microfluidic device to assess human sperm chemotaxis. Using this device, they explore the role of the olfactory receptor OR2H1 in sperm chemotaxis. They show that Methional (activator of OR2H1) may induce a chemotactic response. Some positive aspects deserve recognition being one of them that the authors have made the design of their microfluidic device publicly available and ready to 3D print.
However, I have several concerns that prevent me from accepting this manuscript in its current form.
- The proposed device is an adaptation from the author’s previously published design with only minor modifications (Devigili, et al., Cells 2021, 10, 2472). The novelty lies primarily in its application, not in the platform itself. The authors should mention the novelty of the previous design accordingly.
- The utility of the chamber as a reliable chemotaxis assessment platform is not fully supported by the provided experimental evidence. Here are my arguments:
- The authors state that a chemical gradient forms within 20 seconds based on dye diffusion, but they provide no quantitative data, imaging, or time-course analysis to confirm that the gradient remains stable throughout the 30-minute incubation period used in the accumulation assays.
- The primary outcome measured is the accumulation of cells in the well with the putative chemoattractant. However, the authors do not include essential control conditions needed to distinguish chemotaxis from solely an increase in the linearity without directionality. These controls include uniform (no-gradient) distribution and a reverse-gradient (chemoattractant in the opposite well). Their absence weakens the interpretation of the results as chemotaxis. Since the authors found that Methional increases calcium influx as well as sperm kinematic parameters, the above-mentioned controls would provide strong evidence in favor of chemotactic response.
- The authors do not assess Methional-induced accumulation in a dose-dependent manner. Dose-response experiments were critical in earlier studies on known sperm chemoattractants, such as progesterone and bourgeonal, and should be included here to strengthen the claim.
- The authors relied largely on bioinformatic data from a publicly available database (the Human Protein Atlas) to determine OR2H1 expression. The authors do not experimentally validate the expression of OR2H1 in sperm.
- The mechanistic model in Figure 6 is overly simplistic and omits key steps. It appears to suggest that OR2H1 directly mediates Ca²⁺ influx, which is misleading. The figure lacks detail on GPCR signaling cascades, second messengers, or downstream effectors (e.g., ion channels). In its current form, the model provides little mechanistic insight and may confuse readers.
- The incubation times used to evaluate sperm motility parameters (1 and 3 hours) differ from the 30 minutes used in accumulation experiments. Why did the authors use these different incubation times?
I also have some minor suggestions:
- Please specify the age range of sperm donors, as the manuscript currently states only that they were “young.”
- As I mentioned, the accumulation of sperm in the well with chemoattractant is the main output. Sperm quantification is key to addressing the accumulation rate, thus, the quantification method should be described in detail, including whether dead cells were excluded.
- For the calcium measurements experiments, the number of cells analyzed should be reported.
- DOI for Ref 24 is not correct
In summary, the manuscript presents an interesting application of a microfluidic device to study sperm chemotaxis; however, it lacks key controls and experimental validation. Major revisions are required to support the main conclusions.
Author Response
In this manuscript, from Di Nicuolo and colleagues, the authors present the development of an accessible microfluidic device to assess human sperm chemotaxis. Using this device, they explore the role of the olfactory receptor OR2H1 in sperm chemotaxis. They show that Methional (activator of OR2H1) may induce a chemotactic response. Some positive aspects deserve recognition being one of them that the authors have made the design of their microfluidic device publicly available and ready to 3D print.
However, I have several concerns that prevent me from accepting this manuscript in its current form.
- The proposed device is an adaptation from the author’s previously published design with only minor modifications (Devigili, et al., Cells 2021, 10, 2472). The novelty lies primarily in its application, not in the platform itself. The authors should mention the novelty of the previous design accordingly.
Thank you for the comment. As we mentioned in the paper, we modified the chamber previously described by Devigili et al., with shape and dimensions tailored to human sperm morphology and motility. Given that rapidly motile sperm travel at least 10-25 micrometers per second in a relatively straight path, we estimated a 2.5 cm channel length would allow sperm to accumulate in the recovery channels within 15-30 minutes of incubation. We added this information in the text.
- The utility of the chamber as a reliable chemotaxis assessment platform is not fully supported by the provided experimental evidence. Here are my arguments:
- The authors state that a chemical gradient forms within 20 seconds based on dye diffusion, but they provide no quantitative data, imaging, or time-course analysis to confirm that the gradient remains stable throughout the 30-minute incubation period used in the accumulation assays.
Thank you for the comment. We included a time-course analysis (Supplementary Figure 2) to show the chemical gradient's formation by dye diffusion within 20 seconds and its stability throughout the 30-minute incubation.
2. The primary outcome measured is the accumulation of cells in the well with the putative chemoattractant. However, the authors do not include essential control conditions needed to distinguish chemotaxis from solely an increase in the linearity without directionality. These controls include uniform (no-gradient) distribution and a reverse-gradient (chemoattractant in the opposite well). Their absence weakens the interpretation of the results as chemotaxis. Since the authors found that Methional increases calcium influx as well as sperm kinematic parameters, the above-mentioned controls would provide strong evidence in favor of chemotactic response.
Thanks for your suggestion. To confirm that the observed sperm movement was genuinely due to a chemotactic response and not random motion or other factors, we conducted two control experiments:
- No-gradient control: We performed the experiment without progesterone. As shown in Supplementary Figure 4A, when both wells contained the same solution (e.g., medium without chemoattractant), sperm movement was random and not directed by a chemical gradient.
- Reverse-gradient control: We placed the chemoattractant in the opposite well. Supplementary Figure 4B shows the migrated spermatozoa recovered in the well containing Progesterone (P) relative to the control well (CTR) in this reverse-gradient setup.
3. The authors do not assess Methional-induced accumulation in a dose-dependent manner. Dose-response experiments were critical in earlier studies on known sperm chemoattractants, such as progesterone and bourgeonal, and should be included here to strengthen the claim.
Thanks for your suggestion. To validate our findings, we performed dose-response experiments using Methional at 10 and 100 µM. As Supplementary Figure 5 illustrates, Methional elicited sperm accumulation in a dose-dependent manner, thereby supporting its role as a chemoattractant.
- The authors relied largely on bioinformatic data from a publicly available database (the Human Protein Atlas) to determine OR2H1 expression. The authors do not experimentally validate the expression of OR2H1 in sperm.
Thanks for your comment. We did not experimentally validate OR2H1 expression in human sperm since this data is widely available in the literature. Its expression has, in fact, been confirmed using various techniques, including RNA-Seq and immunofluorescence.
*Flegel, C.; Manteniotis, S.; Osthold, S.; Hatt, H.; Gisselmann, G. Expression Profile of Ectopic Olfactory Receptors De-termined by Deep Sequencing. PLoS ONE 2013, 8, e55368.
*Flegel, C.; Vogel, F.; Hofreuter, A.; Schreiner, B.S.P.; Osthold, S.; Veitinger, S.; Becker, C.; Brockmeyer, N.H.; Muschol, M.; Wennemuth, G.; et al. Characterization of the olfactory receptors expressed in human spermatozoa. Front. Mol. Biosci. 2016, 2, 73. https://doi.org/10.3389/fmolb.2015.00073.
- The mechanistic model in Figure 6 is overly simplistic and omits key steps. It appears to suggest that OR2H1 directly mediates Ca²⁺ influx, which is misleading. The figure lacks detail on GPCR signaling cascades, second messengers, or downstream effectors (e.g., ion channels). In its current form, the model provides little mechanistic insight and may confuse readers.
Thanks for your comment. We modified Figure 6 according to your suggestion.
- The incubation times used to evaluate sperm motility parameters (1 and 3 hours) differ from the 30 minutes used in accumulation experiments. Why did the authors use these different incubation times?
Thanks for your comment. In accumulation experiments, we assess sperm count after 30 minutes of incubation, as the chemical gradient remains stable at this time. For sperm motility parameters (evaluated by CASA), we select 1 and 3-hour time points. These longer durations are more compatible with sperm capacitation, a crucial process during which sperm develop increased motility and prepare for fertilization.
I also have some minor suggestions:
- Please specify the age range of sperm donors, as the manuscript currently states only that they were “young.”
Thanks for your suggestion. We added in the text this information.
- As I mentioned, the accumulation of sperm in the well with chemoattractant is the main output. Sperm quantification is key to addressing the accumulation rate, thus, the quantification method should be described in detail, including whether dead cells were excluded.
Thanks for your comment. We added in the text the information about sperm cells number evaluation (“by Makler Chamber in accordance to the WHO 2021 recommendations”). Moreover, we did not evaluate sperm viability because in all experiments we analyse only motile sperms.
For the calcium measurements experiments, the number of cells analyzed should be reported.
Thanks for your suggestion. We added in the text this information.
- DOI for Ref 24 is not correct
Thanks for your suggestion. We correct the DOI.
In summary, the manuscript presents an interesting application of a microfluidic device to study sperm chemotaxis; however, it lacks key controls and experimental validation. Major revisions are required to support the main conclusions.
Round 2
Reviewer 3 Report
Comments and Suggestions for Authors
The authors have resubmitted the manuscript, addressing my previous concerns. Notably, they performed essential control experiments to validate the proposed device as a reliable tool for evaluating human sperm accumulation via chemotaxis or chemokinesis.
However, I still have one major concern about the mechanistic model presented in Figure 6. Although this version is more refined than the previous one, it lacks proper explanation or discussion from the authors. I understand that proposed signaling pathway may apply to somatic cells expressing OR2H1, but sperm are uniquely specialized. Only a limited number of calcium channels have been validated using electrophysiology. Actually, it was demonstrated that the main calcium channel of the sperm, CatSper, can be directly activated by cAMP or cyclic nucleotide analogs (J. Biol. Chem. (2020)295(38) 13181–13193) or by downstream activation of PKA (J. Biol. Chem. (2018)293(43) 16830–16841). Proposing that a CNG channel is responsible for the increased flagellar beating in response to methional seems like overinterpretation and lacks argumentation.
I strongly suggest that the authors either include a clear discussion highlighting that while CNG channel involvement is hypothesized, it remains speculative; or remove the mechanistic model altogether.
Overall, the manuscript quality is generally good, is well written and open new avenues of research. I consider this manuscript suitable for publication once the authors adequately address this final point.
Author Response
The authors have resubmitted the manuscript, addressing my previous concerns. Notably, they performed essential control experiments to validate the proposed device as a reliable tool for evaluating human sperm accumulation via chemotaxis or chemokinesis.
However, I still have one major concern about the mechanistic model presented in Figure 6. Although this version is more refined than the previous one, it lacks proper explanation or discussion from the authors. I understand that proposed signaling pathway may apply to somatic cells expressing OR2H1, but sperm are uniquely specialized. Only a limited number of calcium channels have been validated using electrophysiology. Actually, it was demonstrated that the main calcium channel of the sperm, CatSper, can be directly activated by cAMP or cyclic nucleotide analogs (J. Biol. Chem. (2020)295(38) 13181–13193) or by downstream activation of PKA (J. Biol. Chem. (2018)293(43) 16830–16841). Proposing that a CNG channel is responsible for the increased flagellar beating in response to methional seems like overinterpretation and lacks argumentation.
I strongly suggest that the authors either include a clear discussion highlighting that while CNG channel involvement is hypothesized, it remains speculative; or remove the mechanistic model altogether.
Overall, the manuscript quality is generally good, is well written and open new avenues of research. I consider this manuscript suitable for publication once the authors adequately address this final point.
- Thanks for your comment. The initially proposed signaling pathway includes the CNG channel, due to its potential relevance in somatic cells expressing OR2H1. While it is well established that CatSper is the primary calcium channel in sperm, its activation by odorants remains unclear. CatSper is known to be directly activated by progesterone and prostaglandins—female-derived factors that stimulate Ca²⁺ influx. Several studies have proposed that ligands such as odorants may regulate calcium entry and sperm motility through G protein-coupled receptors (GPCRs) and cAMP-dependent signaling pathways. Conversely, some evidence suggests that odorants can activate CatSper directly, bypassing GPCR and cAMP pathways (*). To date, only one study has reported that OR1D2 mediates bourgeonal-induced activation of CatSper via a G protein-dependent mechanism (**). Given these findings, we cannot definitively conclude that CatSper is the calcium channel involved. Therefore, we have revised Figure 6 to include CatSper as a potential component of the OR2H1 signaling pathway. We added this point in the Discussion section.
* Brenker, C.; Goodwin, N.; Weyand, I.; Kashikar, N.D.; Naruse, M.; Krähling, M.; Muller, A.; Kaupp, U.M.; Strunker, T. The CatSper channel: A polymodal chemosensor in human sperm. EMBO J. 2012, 31, 1654–1665. https://doi.org/10.1038/emboj.2012.30
** Yi-min Cheng, Tao Luo, Zhen Peng, Hou-yang Chen, Jin Zhang, Xu-Hui Zeng. OR1D2 receptor mediates bourgeonal-induced human CatSper activation in a G-protein dependent manner. bioRxiv, 2019. 757880; doi: https://doi.org/10.1101/757880